# STROKENET: A NEURAL PAINTING ENVIRONMENT

**Ningyuan Zheng, Yifan Jiang & Dingjiang Huang**
School of Data Science and Engineering, East China Normal University
`{10165101164, 10153903133}@stu.ecnu.edu.cn,`
`djhuang@dase.ecnu.edu.cn`

## ABSTRACT

We've seen tremendous success of image generating models these years. Generating images through a neural network is usually pixel-based, which is fundamentally different from how humans create artwork using brushes. To imitate human drawing, interactions between the environment and the agent is required to allow trials. However, the environment is usually non-differentiable, leading to slow convergence and massive computation. In this paper we try to address the discrete nature of software environment with an intermediate, differentiable simulation. We present **StrokeNet**, a novel model where the agent is trained upon a well-crafted neural approximation of the painting environment. With this approach, our agent was able to learn to write characters such as MNIST digits faster than reinforcement learning approaches in an unsupervised manner. Our primary contribution is the neural simulation of a real-world environment. Furthermore, the agent trained with the emulated environment is able to directly transfer its skills to real-world software. [1]

## 1 INTRODUCTION

To learn drawing or writing, a person first observes (encodes) the target image visually and uses a pen or a brush to scribble (decode), to reconstruct the original image. For an experienced painter, he or she foresees the consequences before taking any move, and could choose the optimal action.

Stroke-based image generation is fairly different from traditional image generation problems due to the intermediate rendering program. Raster-based deep learning approaches for image generation allow effective optimization using back-propagation. While for stroke-based approaches, rather than learning to generate the image, it is more of learning to manipulate the painting program.

An intuitive yet potentially effective way to tackle the problem is to first learn this mapping from "stroke data" to the resulting image with a neural network, which is analogous to learning painting experience. An advantage of such a mapping over software is that it provides a continuous transformation. For any painting program, the pixel values of an image are calcuated based on the coordinate points along the trajectory of an action. Specific pixels are indexed by the discrete pixel coordinates, which cuts the gradient flow with respect to the action. In our implementation, the indexing is done by an MLP described in Section 3.

We further define "drawing" by giving a formal definition of "stroke". In our context, a "stroke" consists of color, brush radius, and a sequence of tuples containing the coordinate and pressure of each point along the trajectory. We will later describe this in detail in Section 3.

Based on these ideas, we train a differentiable approximator of our painting software, which we call a "**generator**". We then tested the generator by training a vanilla CNN as an **agent** that encodes the image into "stroke" data as an input for the environment. Our proposed architecture, **StrokeNet**, basically comprises the two components, a **generator** and an **agent**.

Finally, an agent is trained to write and draw pictures of several popular datasets upon the generator. For the MNIST (LeCun & Cortes, 2010) digits, we evaluated the quality of the agent with a classifier trained solely on the original MNIST dataset, and tested the classifier on generated

---

[1]Code for the model at: `https://github.com/vexilligera/strokenet`.

images. We also compared our method with others to show the efficiency. We explored the latent space of the agent as well.

## 2 RELATED WORK

Generative models such as VAEs(Kingma & Welling, 2013; Sohn et al., 2015) and GANs(Goodfellow et al., 2014; Mirza & Osindero, 2014; Radford et al., 2015; Arjovsky et al., 2017) have achieved huge success in image generation in recent years. These models generate images directly to pixel-level and thus could be trained through back-propagation effectively.

To mimic human drawing, attempts have been made by both graphics and machine learning communities. Traditionally, trial-and-error algorithms(Hertzmann, 2003) are designed to optimize stroke placement by minimizing an energy function, incorporating heuristics, e.g., constraining the number of strokes. Concept learning is another example tackling this problem using Bayesian program learning (Lake et al., 2015). Recent deep learning based approaches generally falls into two categories: RNN-based approaches and reinforcement learning.

For RNN-based approaches such as SketchRNN (Ha & Eck, 2017) and handwriting generation with RNN by Graves (Graves, 2013), they both rely on sequential datasets. Thus for unpaired data, those models cannot be applied.

Another popular solution is to adopt reinforcement learning such as "artist agent"(Xie et al., 2012) and SPIRAL (Ganin et al., 2018). These methods train an agent that interact with the painting environment. For reinforcement learning tasks with large, continuous action space like this, the training process can be computationally costly and it could take the agent tens of epochs to converge.

To mitigate this situation, we simulate the environment in a differentiable manner much alike the idea in World Models (Ha & Schmidhuber, 2018; Schmidhuber, 1990; 2018), where an agent learns from a neural network simulated environment. Similar approach is also used in character reconstruction for background denoising(Huang et al., 2018). In our scenario, we train our generator (auto-encoder) by parts for flexible stroke sequence length and image resolution, discussed in Section 3 and 4.

Differentiable rendering is an extensively researched topic in computer graphics. It is used to solve inverse rendering problems. Some differentiable renderers explicitly model the relationship between the parameters and observations (Loper & Black, 2014), others use neural network to approximate the result (Nguyen-Phuoc et al., 2018) since neural nets are powerful function approximators. While little has been done on simulating 2D rendering process adopted in digital painting software, we used a generator neural network to meet our needs.

## 3 STROKENET ARCHITECTURE AND ENVIRONMENT

### 3.1 STROKE

We define a single stroke as follows,

$$s = \{(c,r),(x_1,y_1,p_1),\cdots,(x_n,y_n,p_n)\},\ n = 16, \tag{1}$$

where $c \in \mathbb{R}^3$ stands for RGB color, scalar $r$ for brush radius, and tuple $(x_i,y_i,p_i)$ for an anchor point on the stroke, consisting of $x,y$ coordinate and pressure $p$, and $n$ is the maximum number of points in a single stroke, in this case, 16. These values are normalized such that the coordinates correspond to the default OpenGL coordinate system.

$$c_k,r,p_i \in [0,1],\ x_i,y_i \in [-1,1], \tag{2}$$

for $k = 1,2,3$ and $i = 1,2,\cdots,n$. We used absolute coordinates for each point. It is notable that compared to the QuickDraw (Ha & Eck, 2017) dataset which contains longer lines, our strokes consist of much fewer points. We consider many trajectory points redundant since the stroke lines can be fitted by spline curves with fewer anchor points. For example, to fit a straight line, only two end-points are needed regardless of the length, in other words, stroke curves are usually scale-invariant. However, if we are to sample the data from a user input, we could have dozens of points along the trajectory. Hence we made the assumption of being able to represent curves with a few

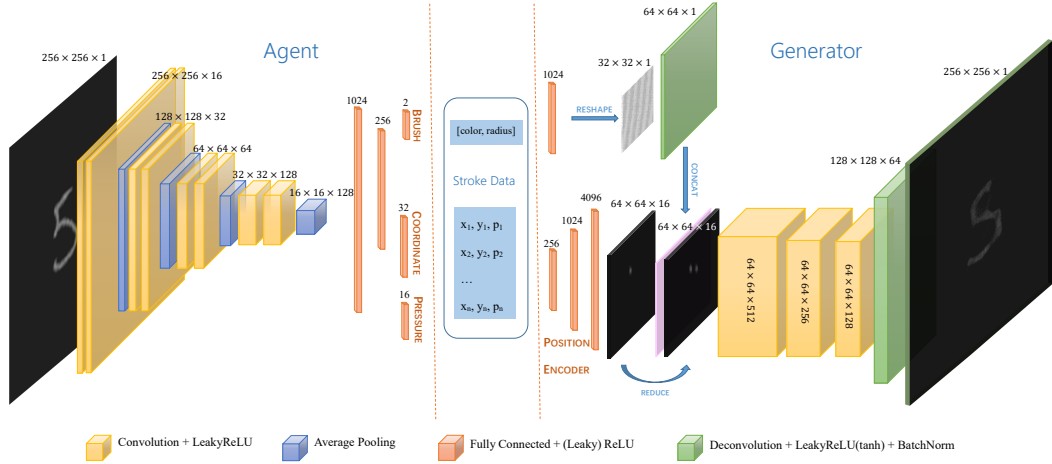

Figure 1: StrokeNet architecture. The generator part of the model outputs $256 \times 256$ images. The position encoder encodes input coordinate into $64 \times 64$ spatial feature for each point. The agent decodes different information about the stroke using three parallel FC-decoders.

anchors. We later showed that a single stroke with only 16 anchors is able to fit most MNIST digits and generate twisted lines in Section 5. We further assumed that longer and more complicated lines can be decomposed into simple segments and extended our experiments to include recurrent drawing of multiple strokes to generate more complex drawings.

## 3.2 GENERATOR

The outline of the StrokeNet architecture is shown in Figure 1. The generator takes $s$ as input, and projects the stroke data with two MLPs. One is the **position encoder** which encodes $(x_i, y_i, p_i)$ into $64 \times 64$ feature maps, the other, **brush encoder** encodes the color and radius of the brush to a single $64 \times 64$ feature map. The color $c'$ is a single gray scale scalar whose value equals to $\frac{1}{3}\sum_{k=1}^{3} c_k$, while color strokes are approximated by channel mixing described in Section 3.4. The features are then concatenated and passed to the (de)convolution layers.

To preserve the sequential and pressure information of each point $(x_i, y_i, p_i)$, the **position encoder** first maps $(x_i, y_i)$ to the corresponding position onto a $64 \times 64$ matrix by putting a bright dot on that point. This is modeled by a 2D Gaussian function with its peak scaled to 1, which simplifies to:

$$g_i(x, y) = \exp[-\frac{1}{2}((x - x_i)^2 + (y - y_i)^2)], \tag{3}$$

for $i = 1, 2, \cdots, n$ where the value is calculated for each point $(x, y)$ on the $64 \times 64$ map. Denote this mapping from $(x_i, y_i)$ to $\mathbb{R}^{64 \times 64}$ as $pos$:

$$m_i = p_i \cdot pos(x_i, y_i), \ m_i \in \mathbb{R}^{64 \times 64}. \tag{4}$$

By multiplying the corresponding pressure $p_i$, we now have $n$ position features, in our setup, sixteen. This part of the generator is trained separately with random coordinates until it generates accurate and reliable signals.

However, if we directly feed these features into the (de)convolutional layers of the network, the generator fails partly due to the sparsity of the single brightness feature. Instead, we take every two neighbouring feature maps and add them together (denoted by "reduce" in Figure 1.),

$$f_i = m_i + m_{i+1}, \ i = 1, 2, \cdots, n - 1. \tag{5}$$

Now, each feature map $f_i$ represents a segment of the stroke. By learning to connect and curve the $n - 1$ "segments", we are able to reconstruct the stroke. By appending the encoded color and radius data we now have the feature with shape $64 \times 64 \times n$. We then feed the features into three (de)convolutional layers with batch-normalization (Ioffe & Szegedy, 2015) activated by LeakyReLU (Xu et al., 2015). The last layer is activated by tanh.

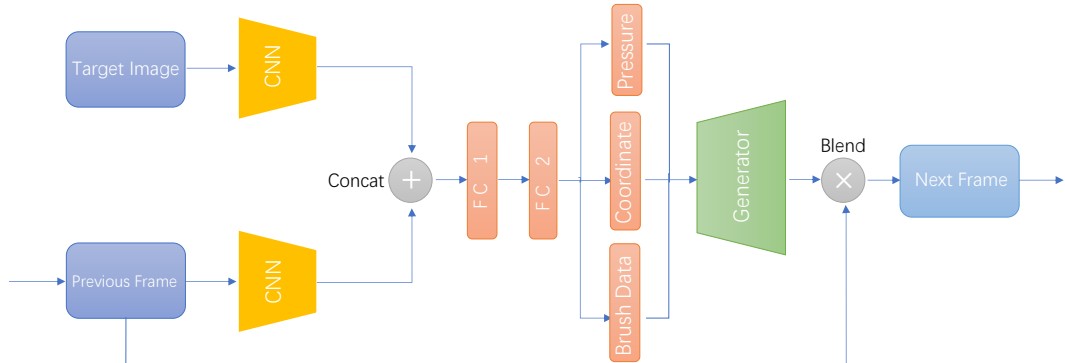

Figure 2: Recurrent version of StrokeNet. Two separate CNNs are used as encoders for the agent.

### 3.3 AGENT

The agent is a VGG (Simonyan & Zisserman, 2014)-like CNN that encodes the target image into its underlying stroke representation $s$. Three parallel FC-decoders with different activations are used to decode position (tanh), pressure (sigmoid) and brush data (sigmoid) from the feature. We used average-pooling instead of max-pooling to improve gradient flow. For the recurrent version of StrokeNet, two separate CNNs are trained for the target image and the drawing frame, as shown in Figure 2. In practice the target image feature is computed once for all steps.

### 3.4 ENVIRONMENT

We first built a painting software using JavaScript and WebGL. We later tailored this web application for our experiment. [2] The spline used to fit the anchor points is centripetal Catmull-Rom (Catmull & Rom, 1974; Barry & Goldman, 1988). A desirable feature about Catmull-Rom spline is that the curve goes through all control points, unlike the more commonly used Bezier curve (Sederberg & Farouki, 1992).

We then interpolate through the sampled points and draw circles around each center point as shown in Figure 3. For each pixel inside a circle, its color depends on various factors including attributes of the brush, blending algorithm, etc. Our generator is trained on the naive brush. When it comes to the color blending of two frames, the generator is fed with the mean value of input RGB color as a gray scale scalar, and its output is treated as an alpha map. Normalization and alpha-blending is then performed to yield the next color frame, to simulate real blending algorithm underlying the software. Denote the generator output at time-step $t$ by $q^{(t)} \in \mathbb{R}^{256 \times 256}$, the frame image by $r^{(t)} \in \mathbb{R}^{3 \times 256 \times 256}$, RGB color of the brush by $c \in \mathbb{R}^3$, the blending process is approximated as follows,

$$n^{(t)} = \frac{q^{(t)}}{\max\limits_{1 \leq i,j \leq 256} q_{ij}^{(t)}}, \tag{6}$$

$$r_k^{(t)} = (J - n^{(t)})r_k^{(t-1)} + c_k n^{(t)} \tag{7}$$

for $k = 1, 2, 3$ corresponding to the RGB channels, where J denotes a $256 \times 256$ all-one matrix.

## 4 TRAINING METHODS

### 4.1 DATASET FOR GENERATOR

For the generator, we synthesize a large amount of samples, each of length $n$. We would like to capture both the randomness and the smoothness of human writing, thus it is natural to incorporate **chaos**, most notably, the motion of three-body (Nielsen et al., 2001).

---

[2] Code for the web application available at: `https://github.com/vexilligera/drawwebapp`.

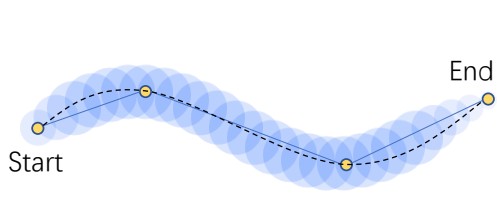

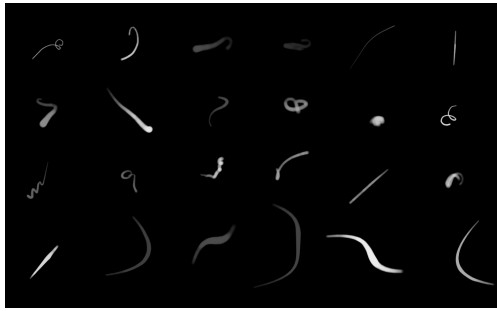

Figure 3: Illustration of how a stroke is rendered.     Figure 4: Images from our three-body dataset.

There is no closed-form solution to three-body problem, and error accumulates in simulation using numerical methods, leading to unpredictable and chaotic results. We simulate three-body motion in space (z-component for pressure) with random initial conditions and sample the trajectories as strokes for our dataset. The simulation is done with a set of equations using Newton's universal law of gravitation:

$$
\begin{cases}
\vec{F_1} = \frac{Gm_1m_2}{\|P_2 - P_1\|_2^3}(P_2 - P_1) + \frac{Gm_1m_3}{\|P_3 - P_1\|_2^3}(P_3 - P_1) \\
\vec{F_2} = \frac{Gm_1m_2}{\|P_1 - P_2\|_2^3}(P_1 - P_2) + \frac{Gm_2m_3}{\|P_3 - P_2\|_2^3}(P_3 - P_2) \,, \\
\vec{F_3} = \frac{Gm_1m_3}{\|P_1 - P_3\|_2^3}(P_1 - P_3) + \frac{Gm_2m_3}{\|P_2 - P_3\|_2^3}(P_2 - P_3)
\end{cases}
\tag{8}
$$

where $P_i(i = 1, 2, 3, \; P_i \in \mathbb{R}^3)$ denotes the position of the three objects respectively, $\vec{F_i}$ denotes the gravitational force exerted on the $i$th object. In our simulation we set mass $m_1 = m_2 = m_3 = 1$ and gravitational constant $G = 5 \times 10^{-5}$. We also always keep our camera (origin point) at the center of the triangle formed by the three objects to maintain relatively stable "footage".

Using this method we collected about 600K images since there is virtually no cost to generate samples. Samples from the dataset are shown in Figure 4.

## 4.2 DATASETS FOR AGENT

To prove the effectivess of our neural environment, we trained an agent to perform drawing task on several popular datasets, from characters to drawings, with the generator part frozen. For MNIST and Omniglot, we trained an agent to draw the characters within one stroke. We later trained the recurrent StrokeNet on more complex datasets like QuickDraw and KanjiVG (Ofusa et al., 2017). We resized all the input images to $256 \times 256$ with anti-alias and paddings.

## 4.3 TRAINING METHODOLOGY

At first we train the **position encoder** guided by function $pos$ that maps a coordinate to a $64 \times 64$ matrix with $l^2$ distance to measure the loss. Next we **freeze** the **position encoder** and train the other parts of the generator, again with $l^2$ loss to measure the performance on the three-body dataset. It can be found that smaller batch size results in more accurate images. We trained the generator with a batch size of 64 until the loss no longer improves. We then set the batch size to 32 to sharpen the neural network.

To train the **agent**, we freeze the **generator**. Denote the agent loss as $l_{agent}$, the generated image and ground-truth image as $i_{gen}$ and $i_{gt}$ respectively, the loss is defined as:

$$
l_{agent} = \|i_{gen} - i_{gt}\|_2^2 + \frac{\lambda}{n-1} \sum_{k=1}^{n-1} \|P_k - P_{k+1}\|_2^2 \,,
\tag{9}
$$

where $P_k = [x_k, y_k, p_k]^T$ is the data describing the $k$th anchor point on the stroke. Here the summation term constrains the average distance between neighbouring points, where $\lambda$ denotes the penalty strength. If we drop this term, the agent fails to learn the correct order of the points in a stroke because the generator itself is, after all, not robust to all cases of input, and is very likely to produce wrong results for sequences with large gaps between neighbouring points.

## 5   EXPERIMENTS

All experiments are conducted on a single NVIDIA Tesla P40 GPU. We first experimented with single step StrokeNet on MNIST and Omniglot, then we experimented recurrent StrokeNet with QuickDraw and Kanji. For the MNIST dataset, we later enforced a Gaussian prior to the latent variable and explored the latent space of the agent by linear interpolation. Finally, for a quantitative evaluation of the model, we trained a classifier on MNIST, and tested the classifier with images generated by the agent. The close accuracies indicate the quality of the generated images.

### 5.1   SINGLE-STEP STROKENET

It can be seen that a single stroke provides rich expressive power for different shapes. The generator generalizes well to unseen stroke patterns other than the synthesized three-body dataset. On the Omniglot dataset, since many characters consist of multiple strokes while the agent can only draw one, the agent tries to capture the contour of the character.

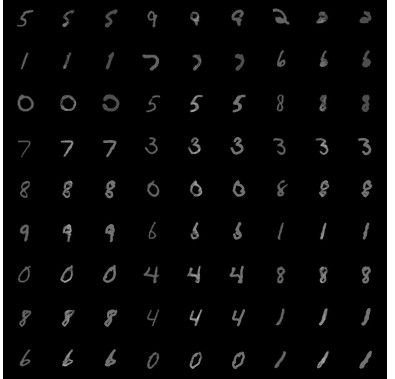

Figure 5: Agent trained on MNIST. MNIST sample (left), generator output (middle), WebApp reconstruction (right). 1.5 epochs.

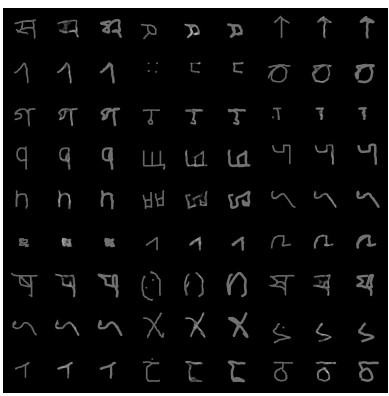

Figure 6: Agent trained on Omniglot dataset learns to "sketch" the characters. Layout is the same as Figure 5. $10^4$ iterations.

### 5.2   RECURRENT-STEP STROKENET

For more complex datasets, multiple steps of strokes are needed. Again the agent does pretty well to capture the contour of the given image. However, it seems that the agent has trouble to recover the details of the pictures, and tends to smear inside the boundaries with thick strokes.

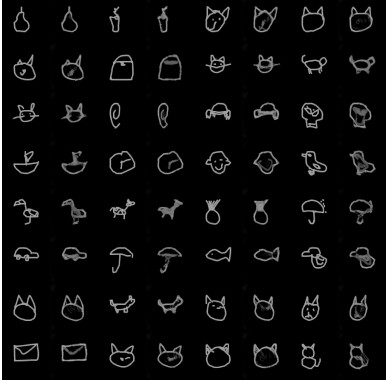

Figure 7: Agent trained on QuickDraw. Sample (left), reconstruction (right). 6 recurrent steps. 300K-image subset, 5 epochs, $10^5$ iterations.

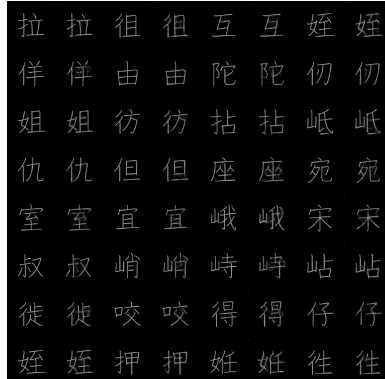

Figure 8: Agent trained on KanjiVG dataset. Evaluated on a test-set of simple characters. $10^5$ iterations. 8 recurrent steps.

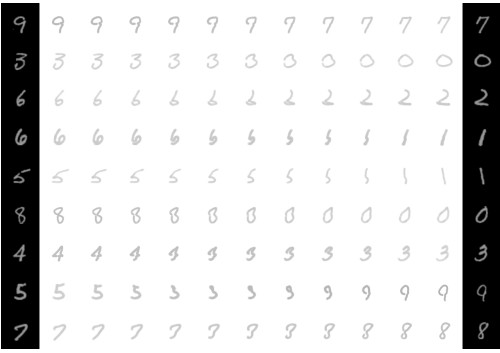

Figure 9: Latent space interpolation. Leftmost and rightmost columns are images from MNIST dataset. Middles are the rendered images with interpolation factors varying from 0 to 1.

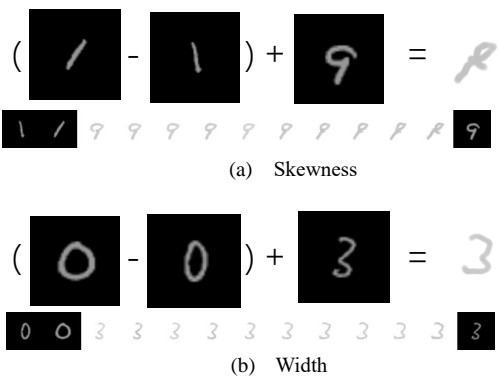

(a) Skewness

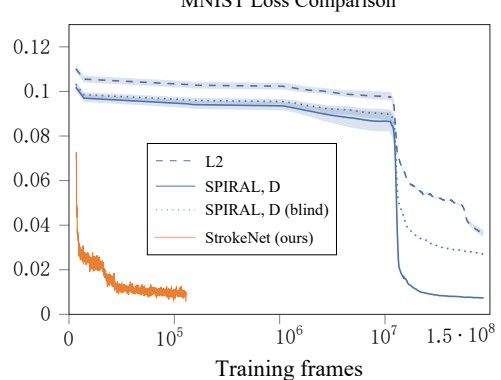

(b) Width

Figure 10: Latent space arithmetics. (a) and (b) demonstrate different attributes of the digits.

## 5.3 LATENT SPACE EXPLORATION

To convert the agent into a latent space generative model, we experimented with the VAE version of the agent, where the feature obtained from the last layer of CNN is projected into two vectors representing the means $\mu$ and standard deviations (activated by softplus) $\sigma$, both of 1024 dimensions. A vector noise of i.i.d. Gaussian $U \sim N(0, I)$ is sampled, latent variable $z$ is given by

$$z = \mu + \sigma \odot U. \tag{10}$$

We did latent space interpolation with the agent trained on MNIST. The simple data led to easily interpretable results. Since the images are generated by strokes, the digits transform smoothly to one another. That is to say, the results looked as if we were directly interpolating the stroke data. Results are shown in Figure 9 and 10.

## 5.4 PERFORMANCE EVALUATION

In order to evaluate the agent, we trained a 5-layer CNN classifier solely on pre-processed MNIST dataset, which is also the input to the MNIST agent. The size of the image is $256 \times 256$, so there is some performance drop to the classification task compared to standard $28 \times 28$ images. The classifier is then used to evaluate the paired test-set image generated by the agent. The accuracies reflect the quality of the generated images. We also compared the $l^2$ loss with SPIRAL on MNIST to illustrate that our method has the advantage of faster convergence over reinforcement learning approaches, shown in Figure 11.

| TEST DATA | ACCURACY |
|---|---|
| Pre-processed images | 90.82% |
| Agent Output (3 steps) | 88.43% |
| Agent Output (1 step) | 79.33% |
| Agent Output (1 step, VAE) | 67.21% |

Table 1: MNIST Classification Accuracies

Figure 11: Comparison of loss curves between StrokeNet (ours) and SPIRAL.

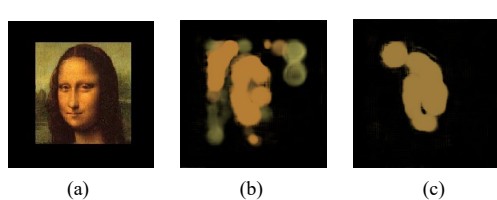

Figure 12: 16-step color image reconstruction. (a) Mona Lisa. (b) Successful reconstruction includes use of different colors. (c) Fail to use different colors.

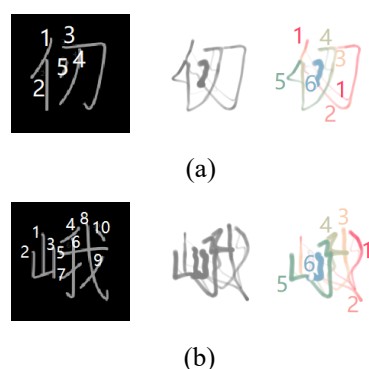

(a)

(b)

Figure 13: Comparison of stroke orders between human and agent. We can see the stroke order is completely chaotic compared to natural order.

## 6 DISCUSSION

For future work, there are several major improvements we want to make both to the network structure and to the algorithm.

The recurrent structure adopted here is of the simplest form. We use this setup because we consider drawing as a Markov process, where the current action only depends on what the agent sees, the target image and the previous frame. More advanced structures like LSTM (Hochreiter & Schmidhuber, 1997) or GRU (Chung et al., 2014) may boost the performance. A stop sign can be also introduced to determine when to stop drawing, which can be useful in character reconstruction. For the agent, various attention mechanism could be incorporated to help the agent focus on undrawn regions, so that smear and blurry scribbles might be prevented.

Secondly, The generator and the agent were trained as two separate parts throughout the experiment. We can somehow train them as a whole: during the training of the agent, store all the intermediate stroke data. After a period of training, sample images from the real environment with the stroke data just collected, and train the generator with the data. By doing so in an iterative manner, the generator could fit better to the current agent and provide more reliable reconstructions, while a changing generator may potentially provide more valuable overall gradients.

It is also found useful to add a bit of randomness to the learning rate. Since different decoders of the agent learn at different rates, stochasticity results in more appealing results. For example, the agent usually fails to generalize to color images because it always sticks with one global average color (as shown in Figure 12). However, it sometimes generates appealing results with some randomness added during the training. As a result of this immobility, the way agent writes is dull compared to humans and reinforcement learning agents like SPIRAL. For instance, when writing the digit "8", the agent is simply writing "3" with endpoints closed. Also, the agent avoids to make intersecting strokes over all datasets, although such actions are harmless and should be totally encouraged and explored! Thus, random sampling techniques could be added to the decision making process to encourage bolder moves. Finally, for the evaluation metrics, the naive $l^2$ loss can be combined with adversarial learning. If paired sequential data is available, we believe adding it to training will also improve the results.

## 7 CONCLUSION

In this paper we bring a proof-of-concept that an agent is able to learn from its neural simulation of an environment. Especially when the environment is deterministic given the action, or contains a huge action space, the proposed approach could be useful. Our primary contribution is that we devised a model-based method to approximate non-differentiable environment with neural network, and the agent trained with our method converges quickly on several datasets. It is able to adapt its skills to real world. Hopefully such approaches can be useful when dealing with more difficult reinforcement learning problems.

## ACKNOWLEDGEMENTS

This work was partially supported by the National Natural Science Foundation of China (U1711262, U1811264, 11501204). We thank our anonymous reviewers for their valuable feedback and opinions.

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

## 8   APPENDIX

### 8.1   ENVIRONMENT DETAILS

Let $P_i = [x_i, y_i]^T$ denote the coordinate of a sampled point. For a curve defined by points $P_0, P_1, P_2, P_3$, the spline can be produced by:

$$C = \frac{t_2 - t}{t_2 - t_1} B_1 + \frac{t - t_1}{t_2 - t_1} B_2, \tag{11}$$

where

$$B_1 = \frac{t_2 - t}{t_2 - t_0} A_1 + \frac{t - t_0}{t_2 - t_0} A_2, \quad B_2 = \frac{t_3 - t}{t_3 - t_1} A_2 + \frac{t - t_1}{t_3 - t_1} A_3,$$

$$A_1 = \frac{t_1 - t}{t_1 - t_0} P_0 + \frac{t - t_0}{t_1 - t_0} P_1, \quad A_2 = \frac{t_2 - t}{t_2 - t_1} P_1 + \frac{t - t_1}{t_2 - t_1} P_2, \tag{12}$$

$$A_3 = \frac{t_3 - t}{t_3 - t_2} P_2 + \frac{t - t_2}{t_3 - t_2} P_3, \quad t_{i+1} = \|P_{i+1} - Pi\|_2^\alpha + t_i.$$

with $\alpha = 0.5$, $t_0 = 0$ and $i = 0, 1, 2, 3$

By interpolating $t$ from $t_1$ to $t_2$ linearly, we generate the curve between $P_1$ and $P_2$. The pressure values between neighbouring points are interpolated linearly.

### 8.2   LOSSES

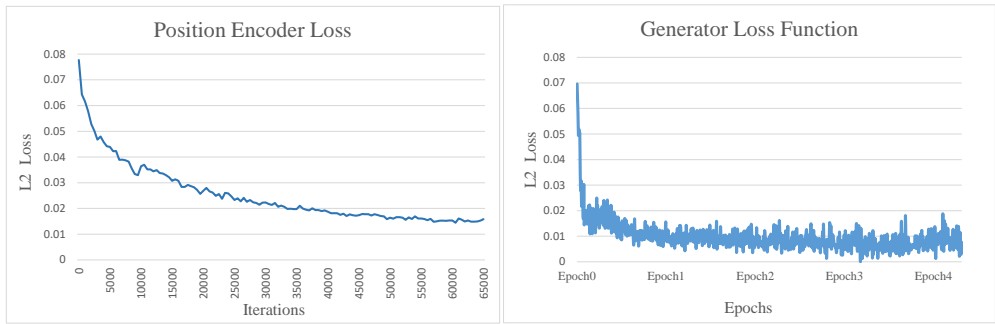

(a) Loss of encoder when trained separately at first.  (b) Loss of generator trained on our 600K dataset.

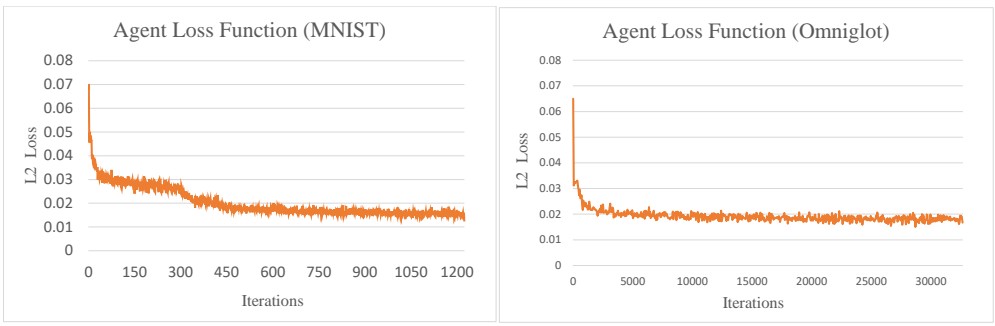

(c) Agent trained on MNIST, $\lambda = 200$.          (d) Agent trained on Omniglot, $\lambda = 200$.

Figure 14:  Training loss of generator and agent. The agent loss equals to the $l^2$ distance between the generator output and agent input plus the penalty term constraining the average point distance within a stroke. For (c) and (d) the learning rate is set to $10^{-4}$, batch size equals to 64.

## 8.3 MISCELLANEOUS RESULTS

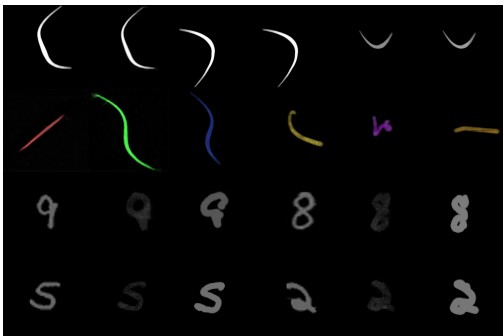

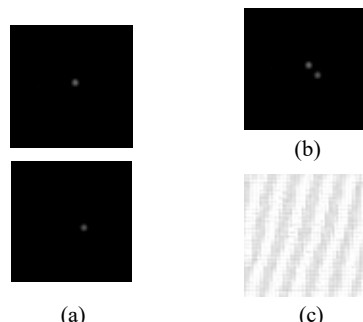

(b)

(a)          (c)

Figure 15: A trained StrokeNet generates images that resemble the output of painting software. The first row depicts results generated by our model (left) and by the software (right) given the same input. The second row shows the model could produce strokes with color and texture using simple arithmetic operations. The third and fourth row shows the model's ability to draw MNIST digits (left) on both its own generative model (middle) and real-world painting software (right).

Figure 16: (a) A trained position encoder maps two $(x_i, y_i, p_i)$ tuples to two feature maps. (b) Each pair of neighbouring features are added together to eliminate sparsity and preserve sequential information. (c) A trained brush encoder encodes color and radius information into a spatial feature which is later concatenated to the end of position features.

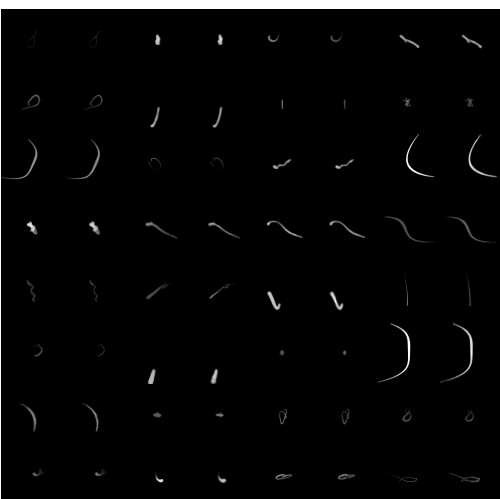

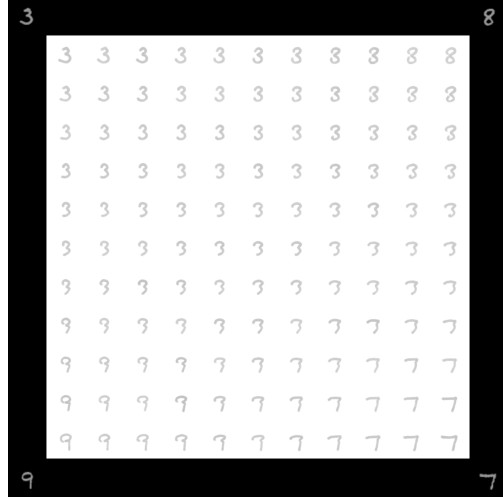

Figure 17: The generator tries to predict what the real environment would ouput given the same input stroke data. Software output (left), generator prediction (right).

Figure 18: Interpolation across four digits. In the corners are the four MNIST samples.

