# OpenReview forum: "StrokeNet: A Neural Painting Environment"
_ICLR.cc/2019/Conference_

### Official Review · AnonReviewer3 · 2018-10-31

**Rating:** 6
**Confidence:** 4

**Review:**

The paper proposes to use a differentiable drawing environment to synthesize images and provides information about some initial experiments.

Not yet great about this paper:
 - the paper feels premature: There is a nice idea, but restricting the drawing environment to be
 - Some of the choices in the paper are a bit surprising, e.g. the lines in the drawing method are restricted to be at most 16 points long. If you look at real drawing data (e.g. the quickdraw dataset: https://quickdraw.withgoogle.com/data) you will find that users draw much longer lines typically.
EDIT: the new version of the paper is much better but still feels like a bit incomplete. I personally would prefer a more complete evaluation and discussion of the proposed method.
 - the entire evaluation of this paper is purely qualitative (and that is not quite very convincing either). I feel it would be important for this paper to add some quantitative measure of quality. E.g. train an MNIST recognizer synthesized data and compare that to a recognizer trained on the original MNIST data.
 - a proper discussion of how the proposed environment is different from the environment proposed by Ganin et al (Deepmind's SPIRAL)

Minor comments:
 - abstract: why is it like "dreaming" -> I do agree with the rest of that statement, but I don't see the connection to dreaming
 - abstract: "upper agent" -> is entirely unclear here.
 - abstract: the footnote at the end of the abstract is at a strange location
 - introduction: and could thus -> and can thus
 - introduction: second paragraph - it would be good to add some citations to this paragraph.
 - resulted image-> resulting image
 - the sentence: "We can generate....data is cheap" - is quite unclear to me at this time. Most of it becomes clearer later in the paper - but I feel it would be good to put this into proper context here (or not mention it)
 - we obtained -> we obtain
 - called a generator -> call a generator
 - the entire last paragraph on the first page is completely unclear to me when reading it here.
 - equations 1, 2: it's unclear whether coordinates are absolute or relative coordinates.
- fig 1: it's very confusing that the generator, that is described first is represented at the right.
 - sec 3.2 - first line: wrong figure reference - you refer to fig 2 - but probably mean fig 1
 - page 3 bottom: by appending the encoded color and radius data we have a feature with shape 64x64xn -> I don't quite see how this is true. The image was 64x64 -> and I don't quite understand why you have a color/radius for each pixel.
 - sec 3.3 - it seem sthat there is a partial sentence missing
 - sec 3.4 - is it relevant to the rest of the paper that the web application exists (and how it was implemented).
 - fig 2 / fig 3: these figures are very hard to read. Maybe inverting the images would help. Also fig 3 has very little value.

---

> ### Author Response · Authors · 2018-11-27
> **Response to ICLR 2019 Conference Paper270 AnonReviewer3**
>
> Thank you for your patient instructions. We followed your advice and made many improvements.
>
> 1.	Indeed, the idea was premature. In the updated version of the paper, we extended the architecture and evaluated our model on various datasets.
>
> 2.	This is a commonly asked question so we added discussion in Section 3.1. We’ve extended our architecture to generate more complex pictures with multiple strokes. Basically, with the power of Catmull-Rom spline, many sampled points in those datasets could be considered redundant. Most strokes we used in writing and drawing are nice and smooth, and we can vectorize them with a few control points and spline algorithms. In other words, those strokes are scale-invariant, so even for long strokes, we can represent them with a few points. In our setup, we found that 16 points offer powerful enough capability to fit various curves.
>
> 3.	We trained a recognizer on the original MNIST dataset and tested it on the generated digits in Section 5.4. The close accuracy reflects the quality of the agent model quantitatively.
>
> 4.	The major difference between our environment and the one used by SPIRAL is that ours uses Catmull-Rom spline while SPIRAL uses Bezier curve. A Bezier curve doesn’t pass through its control points while a Catmull-Rom spline does. Also, the brush rendering algorithm is different depending on what type of brushes the experiments used. From the perspective of training the agent, the nuance between the environment doesn’t affect too much.
>
> Minor points:
> We followed your comments, edited the paper, and moved unnecessary parts to the appendix to avoid confusion.
>
> Regarding the shape of the feature, n points are transformed to n 64x64 feature maps by an MLP, then every neighboring pair of feature maps are added together, which reduces the number of feature maps to n – 1. Then we concatenate the feature of brush data, which is also 64x64, and finally we yield 64x64xn feature maps.
> For the color and radius, we don’t have a color and radius for each pixel, but we do have color and radius for each stroke, as well as the interpolated points along the spline, as shown in Figure 3. For such points and their resulting circle discs, the surrounding pixel values depend on the color and radius.

---

> > ### Comment · AnonReviewer3 · 2018-12-05
> > **Greatly improved.**
> >
> > The authors have addressed most of my comments. I still feel the paper is a bit too early and a more thorough evaluation and explanation would be preferable over the current version.
> > I however feel that the current version is acceptable as is and will adjust my review accordingly.

---

> > > ### Author Response · Authors · 2018-12-11
> > > **Response to Revision**
> > >
> > > Thank you for your revision!
> > >
> > > For future work, we will conduct more complete experiments to provide better evaluation of our model. We will also provide more detailed discussion on our approach in the future version of the paper.

---

### Official Review · AnonReviewer1 · 2018-11-02
**Review of "StrokeNet: A Neural Painting Environment" (Revised, score improvement to 8)**

**Rating:** 8
**Confidence:** 5

**Review:**

Revision:

The authors have taken my advice and addressed my concerns wholeheartedly. It is clear to me that they have taken efforts to make notable progress during the rebuttal period. Summary of their improvements:

- They have extended their methodology to handle multiple strokes
- The model has been converted to a latent-space generative model (similar to Sketch-RNN, where the latent space is from a seq2seq VAE, and SPIRAL where the latent space is used by an adversarial framework)
- They have ran addition experiments on a diverse set of datasets (now includes Kanji and QuickDraw), in addition to omniglot and mnist.
- Newer version is better written, and I like how they are also honest to admit limitations of their model rather than hide them.

I think this work is a great companion to existing work such as Sketch-RNN and SPIRAL. As mentioned in my original review, the main advantage of this is the ability to train with very limited compute resources, due to the model-based learning inspired by model-based RL work (they cited some work on world models). Taking important concepts from various different (sub) research areas and synthesizing them into this nice work should be an inspiration to the broader community. The release of their code to reproduce results of all the experiments will also facilitate future research into this exciting topic of vector-drawing models.

I have revised my score to 8, since I believe this to be at least in the better half of accepted papers at ICLR based on my experience of publishing and attending the conference in the past few years. I hope the other reviewers can have some time to reevaluate the revision.

Original review:

Summary: they propose a differentiable learning algorithm that can output a brush stroke that can approximate a pixel image input, such as MNIST or Omniglot. Unlike sketch-pix2seq[3] (which is a pixel input -> sketch output model based on sketch-rnn[2]), their method trains in an unsupervised manner and does not require paired image/stroke data. They do this via training a "world model" to approximate brush painting software and emulate it. Since this emulation model is differentiable, they can easily train an algorithm to output a stroke to approximate the drawing via back propagation, and avoid using RL and costly compute such in earlier works such as [1].

The main strength of this paper is the original thought that went into it. From reading the paper, my guess is the authors came from a background that is not pure ML research (for instance, they are experts in Javascript, WebGL, and their writing style is easy to read), and it's great to see new ideas into our field. While research from big labs [1] have the advantage of having access to massive compute so that they can run large scale RL experiments to train an agent to "sketch" something that looks like MNIST or Omniglot, the authors probably had limited resources, and had to be more creative to come up with a solution to do the same thing that trains in a couple of hours using a single P40 GPU. Unlike [1] that used an actual software rendering package that is controlled by a stroke-drawing agent, their creative approach here is to train a generator network to learn to approximate a painting package they had built, and then freeze the weights of this generator to efficiently train an agent to draw. The results for MNIST and Omniglot look comparable to [1] but achieved with much fewer resources. I find this work refreshing, and I think it can be potentially much more impactful than [1] since people can actually use it with limited compute resources, and without using RL.

That being said, things are not all rosy, and I feel there are things that need to be done for this work to be ready for publication in a good venue like ICLR. Below are a few of my suggestions that I hope will help the authors improve their work, for either this conference, or if it gets rejected, I encourage the authors to try the next conference with these improvements:

1) multiple strokes, longe strokes. I don't think having a model that can output only a single stroke is scalable to other (simple) datasets such pixel versions of KangiVG [4] or QuickDraw [5]. The authors mentioned the need for an RNN, but couldn't the encoder just output the stroke in a format that contains the pen-down / pen-up event, like the stroke format suggested in [2]? Maybe, maybe not, but in either case, for this work to matter, multiple stroke generation is needed. Most datasets are also longer than 16 points, so you will need to show that your method works for say 80-120 points for this method to be comparable to existing work. If you can't scale up 16 points, would like to see a detailed discussion as to why.

2) While I like this method and approach, to play devil's advocate, what if I simply use an off the shelf bmp-to-svg converter that is fast and efficient (like [6]), and just build a set of stroke data from a dataset of pixel data, and train a sketch-rnn type model described in [3] to convert from pixel to stroke? What does this method offer that my description fails to offer? Would like to see some discussion there.

3) I'll give a hint for as to what I think for (2). I think the value in this method is that it can be converted to a full generative model with latent variables (like a VAE, GAN, sketch-rnn) where you can feed in a random vector (gaussian or uniform), and get a sketch as an output, and do things like interpolate between two sketches. Correct me if I'm wrong, but I don't think the encoder here in the first figure outputs an embedding that has a Gaussian prior (like a VAE), so it fails to be a generative model (check out [1], even that is a latent variable model). I think the model can be easily converted to one though to address this issue, and I strongly encourage the authors to try enforcing a Gaussian prior to an embedding space (that can fit right between the 16x16x128 average pooling op to the fully connected 1024 sized layer), and show results where we can interpolate between two latent variables and see how the vector sketches are interpolated. This has also been done in [2]. If the authors need space, I suggest putting the loss diagrams near the end into the appendix, since those are not too interesting to look at.

4) As mentioned earlier, I would love to see experimental results on [4] KangiVG and [5] QuickDraw datasets, even subsets of them. An interesting result would be to compare the stroke order of this algorithm with the natural stroke order for human doodles / Chinese characters.

Minor points:

a) The figures look like they are bitmap, pixel images, but for a paper advocating stroke/vector images, I recommend exporting the diagrams in SVG format and convert them to PDF so they like crisp in the paper.

b) Write style: There are some terms like "huge" dataset that is subjective and relative. While I'm happy about the writing style of this paper, maybe some reviewers who are more academic types might not like it and have a negative bias against this work. If things don't work out this time, I recommend the authors asking some friends who have published (successfully) at good ML conferences to proof read this paper for content and style.

c) It's great to see that the implementation is open sourced, and put it on github. Next time, I recommend uploading it to an anonymous github profile/repo, although personally (and for the record, in case area chairs are looking), I don't mind at all in this case, and I don't think the author's github address revealed any real identity (I haven't tried digging deeper). Some other reviewers / area chairs might not like to see a github link that is not anonymized though.

So in the end, even though I really like this paper, I can only give a score of 6 (edit: this has since been revised upward to 8). If the authors are able to address points 1-4, please do what you can in the next few weeks and give it your best shot. I'll look at the paper again and will revise the score upwards by a point or two if I think the improvements are there. If not, and this work ends up getting rejected, please consider improving the work later on and submitting to the next venue. Good luck!

[1] SPIRAL https://arxiv.org/abs/1804.01118
[2] sketch-rnn https://arxiv.org/abs/1704.03477
[3] sketch-pix2seq https://arxiv.org/abs/1709.04121
[4] http://kanjivg.tagaini.net/
[5] https://quickdraw.withgoogle.com/data
[6] https://vectormagic.com/

---

> ### Author Response · Authors · 2018-11-27
> **Response to ICLR 2019 Conference Paper270 AnonReviewer1**
>
> Many thanks to your detailed advice and patient review! With your help we made this paper more full-fledged.
>
> As for your major concerns, we made the following improvements:
>
> 1.	We extended the architecture with a simple recurrent structure and implemented a blending algorithm to enable multiple-stroke drawing. We would like to address several issues here:
>
> 1)	Q: “Couldn't the encoder just output the stroke in a format that contains the pen-down / pen-up event, like the stroke format suggested in [2]?”
> A: Indeed. That’s partly what the pressure parameters in the actions are intended for. However, the agent didn’t develop the trick of zero pressure.
>
> 2)	Q: “Why you only allowed 16 points since most datasets contain sequences longer than 16?”
> A: This is a commonly asked question so we added discussion in Section 3.1. Basically, with the power of Catmull-Rom spline, many sampled points in those datasets could be considered redundant. Most strokes we used in writing and drawing are nice and smooth, and we can vectorize them with a few control points and spline algorithms. In other words, those strokes are scale-invariant, so even for long strokes, we can represent them with a few points. In our setup, we found that 16 points offer powerful enough capability to fit various curves.
>
> 2.	This is actually a great idea! However, if we use software like [6] to convert dataset like MNIST to vectors, for digits drawn with thick pen, we would yield the contour of the digits, which is not the sequence how the digit is written. Meanwhile, our agent learns to control the size of the brush to draw digits. There’re limitations to our methods though, discussed in Section 6. Our agent avoids to draw intersecting lines, e.g, when writing “8” it’s actually writing “3” with closed endpoints.
>
> 3.	We added latent space interpolation and latent variable arithmetic for the MNIST agent. We really appreciate this suggestion.
>
> 4.	We added experiments with QuickDraw and KanjiVG using the recurrent version of StrokeNet. For KanjiVG, we found the agent is doodling instead of writing, which resulted in utterly different stroke orders than humans, we compared the stroke orders in Section 6.
>
> For minor points:
> a)	Experiment results are presented in PNG, while diagrams are already exported to PDF.
> b)	We edited most part of the paper so that the language style is more appropriate.
> c)	Next time we will upload the code to anonymous repository. This time, however, the authors made sure in advance so that the github account doesn’t leak any identity information.

---

> ### Author Response · Authors · 2018-12-11
> **Response to Revision**
>
> Thank you very much for your kind revision! Again we really appreciate your constructive suggestions that helped us make this a complete work!

---

### Official Review · AnonReviewer2 · 2018-11-04

**Rating:** 7
**Confidence:** 4

**Review:**

Revision:

The addition of new datasets and the qualitative demonstration of latent space interpolations and algebra are quite convincing. Interpolations from raster-based generative models such as the original VAE tend to be blurry and not semantic. The interpolations in this paper do a good job of demonstrating the usefulness of structure.

The classification metric is reasonable, but there is no comparison with SPIRAL, and only a comparison with ablated versions of the StrokeNet agent. I see no reason why the comparison with SPIRAL was removed for this metric.

Figure 11 does a good job of showing the usefulness of gradients over reinforcement learning, but should have a better x range so that one of the curves doesn't just become a vertical line, which is bad for stylistic reasons.

The writing has improved, but still has stylistic and grammatical issues. A few examples, "there’re", "the network could be more aware of what it’s exactly doing", "discriminator loss given its popularity and mightiness to achieve adversarial learning". A full enumeration would be out of scope of this review. I encourage the authors to iterate more on the writing, and get the paper proofread by more people.

In summary, the paper's quality has significantly improved, but some presentation issues keep it from being a great paper. The idea presented in the paper is however interesting and timely and deserves to be shared with the wider generative models community, which makes me lean towards an accept.

Original Review:

This paper deals with the problem of strokes-based image generation (in contrast to raster-based). The authors define strokes as a list of coordinates and pressure values along with the color and brush radius of a stroke. Then the authors investigate whether an agent can learn to produce the stroke corresponding to a given target image. The authors show that they were able to do so for the MNIST and OMNIGLOT datasets. This is done by first training an encoder-decoder pair of neural networks where the latent variable is the stroke, and the encoder and decoder have specific structure which takes advantage of the known stroke structure of the latent variable.

The paper contains no quantitative evaluation, either with existing methods or with any baselines. No ablations are conducted to understand which techniques provide value and which don't. The paper does present some qualitative examples of rendered strokes but it's not clear whether these are from the training set or an unseen test set. It's not clear whether the model is generalizing or not.

The writing is also very unclear. I had to fill in the blanks a lot. It isn't clear what the objective of the paper is. Why are we generating strokes? What use is the software for rendering images from strokes? Is it differentiable? Apparently not. The authors talk about differentiable rendering engines, but ultimately we learn that a learnt neural network decoder is the differentiable renderer.

To improve this paper and make it acceptable, I recommend the following:

1. Improve the presentation so that it's very clear what's being contributed. Instead of writing the chronological story of what you did, instead you should explain the problem, explain why current solutions are lacking, and then present your own solutions, and then quantify the improvements from your solution.

2. Avoid casual language such as "Reason may be", "The agent is just a plain", "since neural nets are famouse for their ability to approximate all sorts of functions".

3. Show that strokes-based generation enables capabilities that raster-based generation doesn't. For instance, you could show that the agent is able to systematically generalize to very different types of images. I'd also recommend presenting results on datasets more complex than MNIST and OMNIGLOT.

---

> ### Author Response · Authors · 2018-11-27
> **Response to ICLR 2019 Conference Paper270 AnonReviewer2**
>
> Thank you for your reviews and suggestions. We updated the paper with better readability and many clarifications.
>
> Indeed, it’s difficult to provide quantitative analysis and comparisons of this type of generative model considering the limited research done on this topic. In the new version of the paper, we trained a classifier on MNIST to classify generated digits in Section 5.4. The accuracy reflects the quality of the agent. The classifier and the agent are tested using the test-set of MNIST, so neither models have seen the data before. We also added comparison to other methods in that section.
>
> We also added two more datasets to the experiment in Section 5.2, so we can see that the model does have the ability to generalize to different types of data.
>
> As for the differentiability, we also added a discussion in Section 1. In short, when implementing a painting software, we treat the image as a large matrix and index the pixels by integers to calculate new color values for certain pixels. This indexing process is discrete and non-differentiable. While in our neural version of environment, this is done by an MLP, which makes the process differentiable.
>
> For your suggestions, we made the following improvements:
>
> 1.	We edited unnecessary parts to the appendix so that we can explain the problems in greater details. We compared our method with SPIRAL to show improved efficiency in Section 5.4. We trained a recognizer to classify the images generated by our agent to show quantitative results.
>
> 2.	We rewrote many parts of the paper so language is more formal.
>
> 3.	We extended the architecture and experimented with more complex datasets: QuickDraw and KanjiVG, so that we can show the model is able to generalize to different datasets.

---

> ### Author Response · Authors · 2018-12-11
> **Response to Revision**
>
> Thanks again for your constructive advice and kind revision.
>
> Regarding the classification metrics of SPIRAL, we did not have access to SPIRAL generated MNIST test data, and the results presented in their paper weren't enough for evaluation.  We also considered reproducing the SPIRAL experiment, however, since our computation resource was quite limited, training a SPIRAL agent would be virtually impossible. Thus we were only able to compare several ablated models in the experiment. The curves of SPIRAL in Figure 11 is an excerpt from their paper.
>
> As for the writing and Figure 11, after the notification of final acceptance or rejection on Dec. 22, we will update the paper to fix those grammatical and stylistic issues and upload to Arxiv.

---

### Author Response · Authors · 2018-11-27
**Thank you for your reviews!**

We thank our reviewers for their valuable feedback. We’ve updated our paper with several major improvements:
1.	We extended our StrokeNet with a simple recurrent structure, which allowed us to evaluate the model on more complex datasets: QuickDraw and KanjiVG, in Section 5.2.
2.	We trained a classifier on MNIST and tested it on generated digits to provide quantitative analysis of our agent in Section 5.4.
3.	We compared our approach to reinforcement learning approaches like SPIRAL in Section 5.4.
4.	We transformed our agent into a VAE and did latent space interpolation in Section 5.3.
5.	We improved our writing style for better readability.

---

### Public Comment · ~Aaron_Hertzmann1 · 2019-01-01
**Past work in optimizing stroke layout**

I'd suggest looking at/citing past research in optimizing layout of elements in paintings and other forms. Here's a survey paper: http://www.dgp.toronto.edu/~hertzman/sbr02/

RL is one way to try to solve this optimization problem, but it's not necessarily the best, and it's worth at least being aware of past work in this area.

---

> ### Author Response · Authors · 2019-01-01
> **Thank you for your interest and suggestion**
>
> Thank you for your interest and suggestion. We will take a look at the past work and later update the related work section.

---

### Meta-Review · Area_Chair1 · 2018-12-14

**Confidence:** 4
**Recommendation:** Accept (Poster)

**Metareview:**

The paper proposes a novel differential way to output brush strokes, taking a few ideas from model-based learning. The method is efficient in that one can train it in an unsupervised manner and does not require paired data. The strengths of the paper are the qualitative results that demonstrate nice interpolations among other things, on a number of datasets (esp. post-rebuttal).

The weaknesses of the paper are the writing (which I think is relatively easy to improve if the authors make an honest effort) and some of the quantitative evaluation. I would encourage the authors to get in touch with the SPIRAL paper authors in order to get access to the SPIRAL generated MNIST test data and then perhaps the classification metric could be updated.

In summary, from the discussion, the major points of contention were the somewhat lacking initial evaluation (which was fixed to a large extent) and the quality of writing (which could be fixed more). I believe the submission is genuinely novel, interesting (esp. the usage of world model-like techniques) and valuable for the ICLR audience so I recommend acceptance.